# Non-*Saccharomyces* Are Also Forming the Veil of Flor in Sherry Wines

**Marina Ruiz-Muñoz** [1], **María Hernández-Fernández** [1], **Gustavo Cordero-Bueso** [1,*], **Sergio Martínez-Verdugo** [2], **Fernando Pérez** [3] **and Jesús Manuel Cantoral** [1]

1   Área de Microbiología, Departamento de Biomedicina, Biotecnología y Salud Pública, Universidad de Cádiz, Av. República Árabe Saharaui s/n, 11510 Puerto Real, Cadiz, Spain
2   Emilio Lustau S.A., 11401 Jerez de La Frontera, Cadiz, Spain
3   Luis Caballero S.A., 11500 El Puerto de Santa María, Cadiz, Spain
*   Correspondence: gustavo.cordero@uca.es

**Abstract:** Biological ageing is an essential process for obtaining some distinctive Sherry wines, such as Fino and Manzanilla. It occurs after the fermentation of the grape must due to the appearance of a biofilm on the surface of the wine called "veil of flor". Yeasts belonging to the *Saccharomyces cerevisiae* species mainly comprise such biofilm. Although other species have also been found, these have been traditionally considered spoilage. Indeed, it has even been hypothesised that they may not be able to form biofilm on their own under such conditions. In the present work, four different non-*Saccharomyces* yeasts isolated from barrels in the Jerez area under biological ageing have been characterised through their physiological abilities, including extracellular enzymatic and biofilm-forming capabilities. Results showed not only a surprising ethanol tolerance, above 15.5% in all cases, but also a significant degree of extracellular enzyme production, highlighting the urease and proteolytic activities found in *Pichia manshurica,* as well as lipolytic activity in *Pichia kudriavzevii*, *Pichia membranifaciens* and *Wicherhamomyces anomalus*. As a conclusion, these non-*Saccharomyces* could be very interesting in the oenological field, beyond improving the organoleptic characteristics as well as technological features in these wines.

**Keywords:** non-*Saccharomyces*; *Pichia* spp.; Sherry wines; biofilm; metabolic properties; extracellular enzymes; biological ageing

## 1. Introduction

Biological ageing in Sherry wines (i.e., Fino in Jerez and Manzanilla in Sanlúcar de Barrameda areas, Spain) occurs after alcoholic fermentation of the grape must from *Vitis vinifera* L. var. Palomino Fino, and subsequent fortification with wine alcohol to 15–15.5% (*v*/*v*), which is then called "sobretablas". Once fortified, a biofilm naturally appears on the surface of the wine, triggering this process by means of a dynamic ageing system, in which all the ageing scales are blended from earlier vintages, the last scale ("solera") being the one considered as a finished Sherry wine [1].

It is well-established that this biofilm, also known as veil of flor, consists mostly of yeasts belonging to the *Saccharomyces cerevisiae* species [2,3], although others have also been found [4]. Its formation is prompted by the lack of readily assimilable metabolites, whereby flor yeasts modify their metabolism to aerobically assimilate other carbon sources such as ethanol or glycerol. Other compounds are then obtained, such as acetaldehyde, acetoin or higher alcohols, which are considered distinctive and desirable in this type of wine. In the case of Sherry wine, the presence of yeasts other than *Saccharomyces*, belonging mostly to the genus *Pichia*, had already been reported from earlier studies [3,5,6], although it was not possible to go further, neither at the species level, due to the inherent limitations of the existing identification techniques.

Traditionally, several efforts have been made to limit the presence of non-*Saccharomyces* in winemaking due to the belief that certain species could produce undesirable compounds [7]. They were probably perceived as being responsible for these problems due to their isolation from spoiled wines [8], as well as their lower frequency with respect to the main *S. cerevisiae* species and their very different capabilities compared to the usual alcoholic fermentation [9]. This is the case for some *Pichia* strains, which have been identified as causing wine spoilage [10,11]. However, non-*Saccharomyces* have been increasingly considered, not only to improve the organoleptic characteristics of wine, but also to solve technological problems skills [12]. Therefore, the overall oenological effects of these non-*Saccharomyces*, including *Pichia* spp. [13], have been reviewed to better assess their properties and benefits in the wine industry. In this sense, due to the wide range of non-*Saccharomyces* hydrolytic capabilities, they can provide a higher release of metabolites during biological ageing, such as polysaccharides, as well as promote a higher degree of autolysis of dead yeast cells [14]. They may also promote the inhibition of *Brettanomyces* spp. growth, one of the main causes of spoilage in Sherry wines, by producing potent killer toxins [15,16]. In addition, they could also contribute to the safety of these wines, e.g., through the reduction of the ethyl carbamate content, either by the reduction of its precursors, such as urea or citrulline [17], or via the production of enzymes capable of degrading it [18]. It is also known that gene expression and related phenotypic features in some microorganisms can occur under specific and stressful conditions, even leading to morphological change and biofilm formation [19,20]. In this particular case, where the environmental conditions are highly stressful, it seems likely that this could become a highly selective medium for yeasts to develop capabilities that could be very interesting in different oenological fields. They generally present low fermentation yields, and are more sensitive to ethanol stress [21], but may display a great range of possibilities for fermentations to provide distinctive aromas and flavours.

To the best of our knowledge, little is known about the role in forming the veil of flor and the metabolic peculiarities of these non-*Saccharomyces*. Thus, the aim of this work was to identify and characterise native non-*Saccharomyces* strains isolated in Sherry wines from the Jerez area during biological ageing.

## 2. Materials and Methods

### 2.1. Yeast Strains Origin and Culture Media

Yeasts evaluated in the present work were isolated during an intensive sampling carried out during the years 2017–2019 in three wineries producing Fino wine in the Jerez region using the same base wine, although three different wines are finally obtained [22]. Identification was carried out using the 5.8S internal transcribed spacer (ITS) rRNA region amplification, and its restriction through *Hinf* I, *Cfo* I and *Hae* III endonucleases [23]. The amplified products and their restriction fragments were analysed on 2.5% (*w/v*) agarose gels (Condalab, Madrid, Spain) in 1× EDTA buffer (Tris 89 mM, boric acid 89 mM and EDTA 2 mM, pH 8.3) using GeneRuler 100 bp Plus Ladder (Thermo Fisher Scientific, Waltham, MA, USA) as standard. Gels were visualised on a TVC312 UV transilluminator (Bio-Rad, Hercules, CA, USA) after staining in ethidium bromide (10 μL/mL, Sigma-Aldrich, St. Louis, MO, USA).

Species identification was confirmed via the sequencing of the 5.8S rRNA using primers ITS1 and ITS4 as well as the D1/D2 domain of the 26S rRNA in EZ-sequencing using the primers NL1 and NL4 [24] (Macrogen Inc., Seoul, Korea). Sequences were identified via BLAST and compared with European Molecular Biology Laboratory (EMBL) databases. An identity ≥98% and a query cover ≥90% were the thresholds set to correctly identify these yeasts. Besides *S. cerevisiae*, four different species were found: *Pichia kudriavzevii*, *Pichia manshurica*, *Pichia membranifaciens* and *Wickerhamomyces anomalus* (accession numbers MT043929, MT043930, MT043927 and MT043928, respectively). Once identified, the isolates belonging to each species were analysed using RAPD-PCR with the M13 primer [25]. In this case, the obtained products were analysed on 1.5% (*w/v*) agarose gels.

The yeast strain used as a control in the experiments carried out belongs to the species *S. cerevisiae*. It was isolated from the same samplings, being the predominant strain in the biofilms analysed. It was characterised in a previous study carried out by the same research group [22], in which it was called ScA.

The media used were for routine culture were YPD (1% yeast extract, 2% peptone, 2% glucose and 2% agar if necessary), WL Nutrient Agar (Oxoid$^{TM}$) and BBL CHROMagar Candida medium (BD Diagnostics).

### 2.2. Physiological Characterisation

#### 2.2.1. Fermentation and Assimilation of Different Carbon and Nitrogen Sources

Biochemical tests were carried out to assess the capabilities of fermentation and assimilation of different carbon and nitrogen sources (specifically, glucose, galactose, sucrose, maltose, starch, cellobiose, lactose, raffinose, trehalose, inulin, meliobiose, xylose, ammonium citrate, nitrate, nitrite, glucosamine and urea) as stated by Kurtzman et al. [26]. Tests were carried out in 96-well polystyrene plates, and a plate reader Nunc™ 96-well (Thermo Fisher Scientific, MA, USA) was used to measure the optical density (OD) at 620 nm every hour for 120 h, as previously described [27]. The kit RapID$^{TM}$ Yeast Plus System (Remel, San Diego, CA, USA) was also used following the manufacturer's instructions.

#### 2.2.2. Ethanol Resistance

Ethanol resistance was evaluated as described by Aranda et al. [28] with some modifications. Yeast cultures reached their exponential phase for 24 h at 28 °C in YPD. Afterwards, wine ethanol was added to obtain final concentrations ranging between 13 and 19% (*v/v*) in YEP medium (1% yeast extract, 2% bacteriological peptone, pH 4.5). These media were inoculated with the cell suspensions up to an OD$_{600}$ of 0.3. The ability of the yeast isolates to grow was automatically determined via OD$_{600}$ using a Nunc™ 96-well plate reader (Thermo Fisher Scientific, MA, USA) every hour for 120 h to obtain the growth curves in each case. After that, serial dilutions were spotted into YPD plates and incubated at 28 °C for 48 h until colonies appeared, and were counted to numerically confirm what was observed.

#### 2.2.3. Determination of Extracellular Enzymatic Activities

#### β-Glucosidase Activity

Extracellular β-glucosidase activity was determined following the method proposed by Gaensly et al. [29], based on the use of arbutin (glucosylated hydroquinone), peptone as a nitrogen source and ferric ammonium citrate solution, together with the capacity for growth in plates, with cellobiose as a unique carbon source (0.67% YNB, 1% cellobiose, 2% agar, pH 5.5). A positive result was indicated by growth and browning of the media.

#### Cellulase Activity

Cellulase activity was assessed as stated [30] on plates of Yeast Nitrogen Base medium (YNB, Difco$^{TM}$, Saint Ferréol, France) with 0.4% (*w/v*) sodium carboxymethyl cellulose as the sole carbon source and 2% (*w/v*) agar at pH 5.5. Plates were incubated for seven days at 28 °C, and then coated with a 0.5% (*w/v*) Congo red solution for 30 min, rinsed with water and washed twice with a 1 M NaCl solution. Cellulase activity was considered being positive when a yellow halo appeared around the colony on a red background.

#### Protease Activity

Extracellular protease production was tested on skimmed milk agar plates as described by Mangunwardoyo et al. [30]. The presence of clear halos around the colonies after 5 days incubation at 28 °C was considered as a positive reaction.

Lipolytic Activity

The lypolytic activity was analysed by using a previously described medium [31], which contains 1% (*w/v*) bacteriological peptone, 0.5% (*w/v*) NaCl and 0.01% (*w/v*) CaCl$_2$ · 1 H$_2$O, 2% (*w/v*) agar at pH 7.4. After sterilisation, a final aqueous concentration of 1% (*v/v*) Tween 20 was added to the medium. The presence of halos around yeast colonies after incubation for 5 days at 28 °C was indicative of lipolytic activity.

Urease Activity

Regarding urease activity, it was performed in slant tubes containing Yeast Carbon Base medium (YCB, Difco$^{TM}$, Saint Ferréol, France) with 0.12% (*w/w*) phenol red and 1.5% agar, and supplemented with a urea solution (Labkem, Barcelona, Spain) to a final concentration of 2% (*v/v*) at pH 6.5. The presence of a colour shift from yellow to orange-red after yeast growth during 5 days at 28 °C was considered urease positive.

### 2.3. Biofilm-Forming Characterisation

Firstly, the ability of non-*Saccharomyces* yeasts to form a biofilm was analysed in both a 15.5% (*v/v*) base wine fortified with wine ethanol and a synthetic growth-promoting biofilm medium proposed by Moreno-García et al. [32], with some modifications (1% *v/v* glycerol, 4.5% *w/v* glutamic acid, 12% *v/v* ethanol and 13% *w/v* ammonium sulphate). Erlenmeyer flasks containing 500 mL of such medium were inoculated with 10$^6$ cell/mL, and were incubated in static at 25 °C. Biofilm formation was observed daily until the surface was covered (20 days maximum).

Cellular MAT formation was also evaluated to determine the production of large colonies on the surface of low-density agar plates as described [33], using 55 mm diameter petri plates with agar 0.3% (*w/w*). Plates were incubated at 25 °C over 5 days, observing colony growth in such medium.

An adhesion assay on polystyrene was performed as previously described by Zara et al. [34] with slight modifications. Exponential-phase cultures were prepared via inoculation of yeast pre-cultures in YPD at 30 °C overnight. Cells were collected and grown in 15 mL YNB medium supplemented with 2% glucose at 30 °C under shaking conditions (180 rpm). Within 12 h, cells were harvested and inoculated to YNB medium supplemented with 0.1% glucose to an OD$_{600}$ 1.0. Then, 100 μL of the culture were inoculated into individual wells of polystyrene 96-well plates and incubated for 12 h in static at 30 °C to promote cell adhesion. An equal volume of 1% *v/v* crystal violet solution (Panreac, Barcelona, Spain) was added, incubated over 20 min and washed by rising the wells repeatedly with sterile distilled water. After dabbing with absorbent paper, 100 μL of 0.1% SDS (Panreac, Barcelona, Spain) was added to each well to solubilise the crystal violet for 30 min at room temperature. All samples were transferred to a new microplate and absorbance was measured at 570 and 590 nm.

Cell surface hydrophobicity is another characteristic related to biofilm formation, which was evaluated following the method previously proposed by Silva-Dias et al. [28] through the two-phase water-hydrocarbon method. Briefly, a suspension of yeast cells was inoculated into YNB at 25 °C under shaking conditions (180 rpm) for 24 h. Cultures were then diluted with an equal volume of YNB medium, and subsequently, the OD$_{600}$ was measured (D$_0$). After 90 min, 400 μL of octane was added, vortexed for 3 min and rested for 1 min to determine the OD$_{600}$ of the aqueous phase (D$_1$). The hydrophobicity percentage was calculated as:

$$\% \text{ Hydrophobicity} = (D_0 - D_1)/D_0 \times 100 \qquad (1)$$

### 2.4. Scanning Electron Microscopy (SEM)

Sample preparation was performed on biofilm-forming cultures incubated under static conditions at 25 °C in a base wine fortified to 15.5% (*v/v*) with wine ethanol. For the sampling and processing of yeast biofilms, 22 × 22 mm coverslips dipped in poly-L-lysine

and were used once dried [35]. To fix the samples, 2.5% glutaraldehyde in 0.1 M sodium cacodylate buffer, pH 7.4, was used for 2 h at 4 °C. Subsequently, the samples were washed three times in 0.1 M sodium cacodylate buffer at 15 min intervals, followed by dehydration with increasing concentrations of acetone (70%, 90% and 100%) for 30 min each at 4 °C. After carrying out the critical point and drying overnight in a desiccator, the biofilms were metallised in gold and examined using a field-emission scanning electron microscope (FE-SEM) Nova NanoSEM 450.

### 2.5. Statistical Analysis

Three independent replicates of three biological samples was assessed for each analysis. The arithmetic mean, as well as the standard deviation, was calculated. Means were compared by using Student's *t*-test (one-way analysis of variance (ANOVA) or Welch test, as appropriate), once the normality of the data was tested by using the Shapiro–Wilks test. Subsequently, Tukey's HSD post hoc correction test was applied. A *p*-value < 0.05 was considered statistically significant.

A phylogenetic tree was inferred using the online software Phylogeny [36], using in the PhyML the maximum likelihood with aLRT program. The tree obtained was analysed using TreeDyn software, with the input data in Newick Format.

## 3. Results

### 3.1. Characterisation of Non-Saccharomyces Strains

Non-*Saccharomyces* yeasts isolated during the previous exhaustive sampling performed by Ruiz-Muñoz et al. [22] were spotted on chromogenic media (both WL and Chromeagar) to analyse whether any differences could be observed between them. Since the strains identified within each species showed a similar phenotype and growth form, we tried to evaluate their intraspecific diversity via amplification with the M13 primer, but no differences were obtained between them (data not shown). It was therefore assumed that one biotype of each species was being worked with, although five isolates were taken randomly from each species to evaluate their properties.

Furthermore, the BLAST comparisons were confusing in the case of *P. manshurica*, since in no case was an identity rate higher than 98% achieved. Therefore, the ITS1–ITS4 sequences of the strains isolated in this work, together with the type strains of each of them in the CBS, were aligned, and a phylogenetic tree was made (Figure 1).

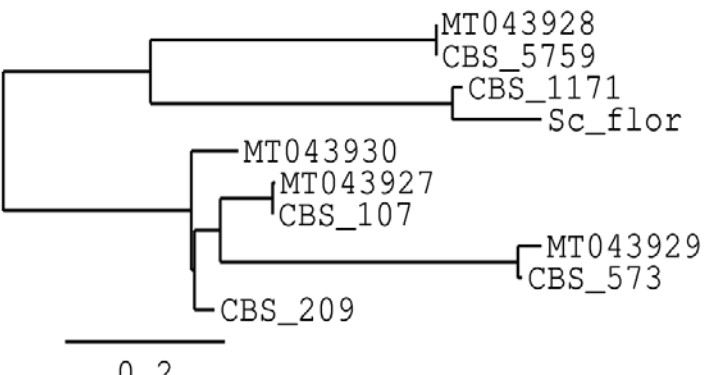

**Figure 1.** Phylogenetic tree with the non-*Saccharomyces* isolated in the present work based on their ITS1–ITS4 regions, as well as the sequences of the subsequent type strain and *S. cerevisiae* flor. MT043927 to MT043930 are the ITS regions for the non-*Saccharomyces* isolates (*P. membranifaciens*, *W. anomalus*, *P. kudriavzevii* and *P. manshurica*, respectively), while *Sc* flor corresponds to the ITS region of the *S. cerevisiae* flor used as a control; CBS107, CBS5759, CBS573, CBS209 and CBS1171 are the accession numbers for the ITS region of the type strain for each species, respectively.

As expected, the species analysed, including *S. cerevisiae*, were very close to each other phylogenetically. However, the *P. manshurica* strain is clearly closer to the rest of the flor,

especially for *P. membranifaciens* and *P. kudriavzevii*, than the sequence of the type strain deposited in the CBS for this species.

The ability of the selected strains to perform both fermentation and assimilation of different carbon and nitrogen sources was evaluated, as shown in Table 1. These strains, isolated in this particular environment, showed a higher capacity to ferment, but especially to assimilate, different carbon and nitrogen sources, including some that are highly desirable during biological ageing. These capabilities seem to be much higher than previously reported in such species [26]. Specifically, the ability to assimilate ethanol (in both *P. manshurica* and *P. kudriavzevii* strains) and glycerol (in the four species studied) is especially remarkable, as these are two of the major compounds used as a carbon source during the biological ageing of Sherry wines.

**Table 1.** Fermentation, assimilation and extracellular enzymatic activities of selected non-*Saccharomyces* yeasts. Values are expressed as (+) if strains showed the specific capability, and (−) if not.

| | | *P. manshurica* | *P. membranifaciens* | *P. kudriavzevii* | *W. anomalus* |
|---|---|:---:|:---:|:---:|:---:|
| **Fermentation** | **Glucose** | + | + | − | + |
| | **Galactose** | + | − | + | + |
| | **Maltose** | − | − | + | − |
| | **Sucrose** | + | − | − | + |
| **Assimilation** | **Glucose** | + | + | + | − |
| | **Ethanol** | + | − | + | − |
| | **Glycerol** | + | + | + | + |
| | **Citrate** | + | − | + | − |
| | **Nitrite** | + | − | + | − |
| | **Urea** | + | − | − | − |
| **Extracellular activities** | **Lipolytic** | + | − | + | + |
| | **Proteolytic** | + | − | − | − |
| | **Cellulolytic** | − | + | − | + |
| | **Urease** | + | − | − | − |
| | **β-glucosidase** | − | − | + | + |

In addition, some of the enzymatic activities considered as being most important in the biological ageing process were also evaluated. Among them, the lipolytic and proteolytic capacity of the *P. manshurica* strain was surprising, as well as its urease capacity. The *P. kudriavzevii* strain showed an interesting lipolytic and glucosidase capacity. On the other hand, strains belonging to the species *P. membranifaciens* and *W. anomalus* showed cellulolytic capacity, while *W. anomalus* also showed β-glucosidase and lipolytic activity.

### 3.2. Biofilm-Forming Evaluation

As shown in Table 2, from all analyses performed, the four non-*Saccharomyces* strains were able to form biofilms at least as efficiently as the control *S. cerevisiae* flor. Hence, they showed high ethanol resistance and a relatively high hydrophobicity, as well as adhesion to plastics and cellular MAT formation.

All four yeast strains analysed in the present work showed a resistance to ethanol above 15.5% (*v/v*), reaching 17% in the case of the *P. manshurica* strain. The quickest to form a biofilm was *P. manshurica* (at 7 days in base wine fortified to 15.5% of ethanol), followed by *W. anomalus* and *P. kudriavzevii* (11 days). The *P. membranifaciens* strain took the longest time to form a biofilm (12 days), showing on the other hand the highest adhesion capacity (Table 2). The hydrophobicity found was also relatively high, above 85% in all cases. Moreover, all non-*Saccharomyces* showed a mature MAT within 5 days. Macroscopic observation revealed that they were composed of a central core and a more or less rough surface with more or less serrated edges (Figure 2). Major differences in diameter were detected, with *P. membranifaciens* showing the largest calibre and *S. cerevisiae* the smallest.

**Table 2.** Biofilm-forming properties (ethanol resistance (%, *v/v*), biofilm (days until formation), adhesion to polystyrene (differences in absorbance at 570 nm) and hydrophobicity (%)) of the four non-*Saccharomyces* yeasts plus the *S. cerevisiae* flor used as a control. The characters [a], [b], and [c] mean significant differences at $p \leq 0.05$, according to Tukey's test.

| | Biofilm Formation (Days) | Hydrophobicity (%) | Ethanol Resistance (%) | Adhesion (ΔAbs570) |
|---|---|---|---|---|
| *P. manshurica* | 7 ± 0.22 [c] | 93 ± 1.25 [b] | 17.00 ± 0.25 [a] | 6.07 ± 0.23 [c] |
| *P. membranifaciens* | 13 ± 0.42 [a] | 96 ± 3.20 [a] | 15.50 ± 0.55 [b] | 10.32 ± 0.18 [a] |
| *P. kudriavzevii* | 11 ± 0.31 [b] | 94 ± 2.20 [ab] | 16.00 ± 0.60 [ab] | 8.72 ± 0.13 [b] |
| *W. anomalus* | 9 ± 0.55 [bc] | 88 ± 2.65 [c] | 16.50 ± 0.20 [a] | 8.82 ± 0.09 [b] |
| *S. cerevisiae*flor | 12 ± 0.34 [b] | 95 ± 1.85 [a] | 16.00 ± 0.30 [ab] | 7.78 ± 0.11 [ab] |

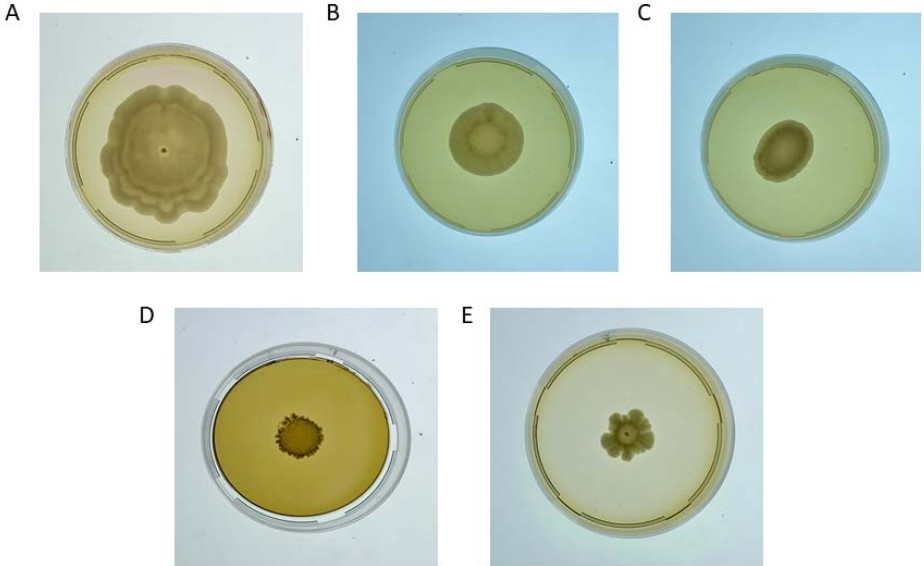

**Figure 2.** MAT morphology on YPD with low-density agar (0.3%, *w/w*): (**A**) *P. membranifaciens*; (**B**) *P. kudriavzevii*; (**C**) *P. manshurica*; (**D**) *W. anomalus*; (**E**) *S. cerevisiae* flor.

Furthermore, the strains analysed were able to form biofilm on their own in pure culture and in a similar way to the *S. cerevisiae* strain used as a control in base wine; that is, wine fortified with up to 15.5% of ethanol. These biofilms were not only observed macroscopically, but also via scanning electron microscopy (Figure 3). Many different morphologies and structures were observed in the biofilms developed by each non-*Saccharomyces* yeast.

Regarding the biofilm formed by *W. anomalus* (Figure 3A), it seemed to be the thinnest and weakest one, showing at a microscopic level an insufficiently consistent matrix between the yeasts, and it appears that their interaction is mainly via adhesins. This observation can also be extended to the biofilm formed by *P. kudriavzevii* (Figure 3B), where despite having a relatively thicker biofilm, the adhesin binding is even more evident.

In the case of *P. manshurica* (Figure 3C), a dense, homogenous biofilm was formed. A very tight network of yeast was observed, with the yeast cells embedded in an extracellular matrix, which seemed to start developing from the newly budding yeast cells.

Except for *P. kudriavzevii*, the biofilm formed for these non-*Saccharomyces* yeasts had a specific three-dimensional structure, different from each other. The most compact and rough-looking biofilm, i.e., most similar to that formed by *S. cerevisiae*, was developed by *P. membranifaciens* (Figure 3D). Microscopically, the matrix did not seem to have as well-defined a three-dimensional structure as in the others, but the cells seemed to be strongly attached between them.

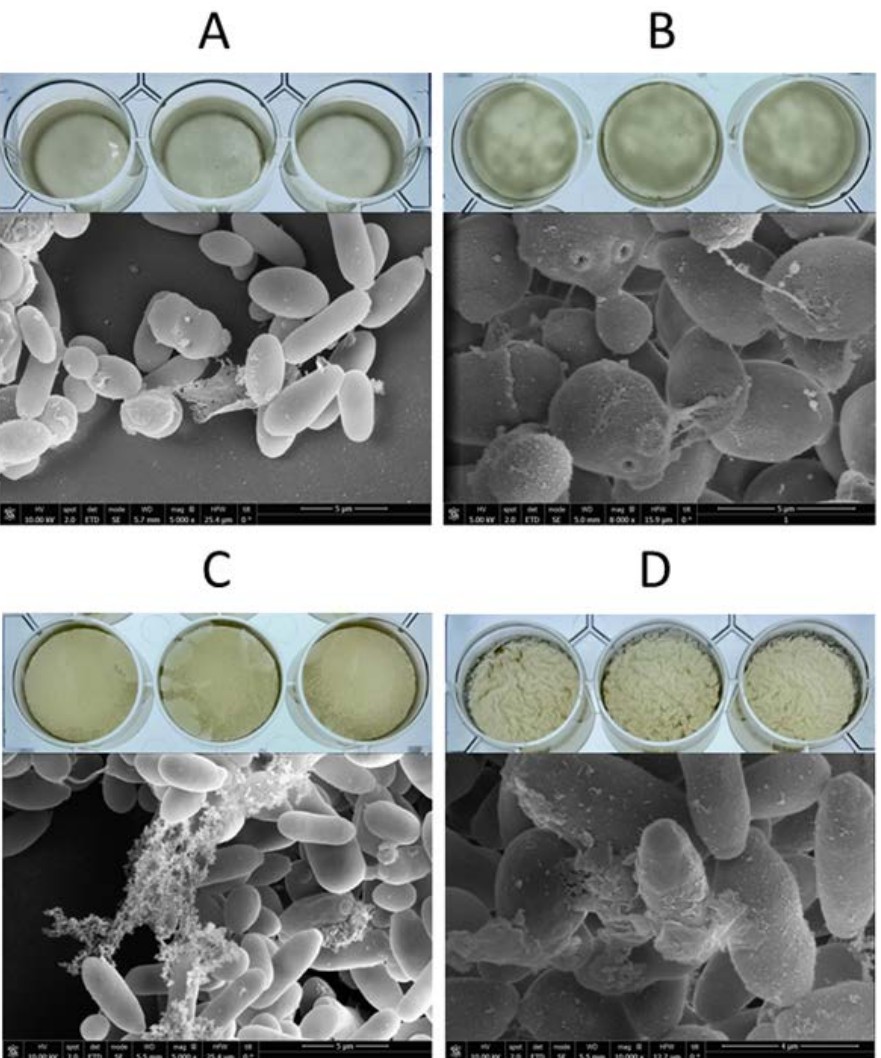

**Figure 3.** Flor yeast biofilm at a macroscopic level in base wine, and scanning electron microscopy of each biofilm formed by the non-*Saccharomyces* strains analysed: (**A**) *W. anomalus*; (**B**) *P. kudriavzevii*; (**C**) *P. manshurica*; (**D**) *P. membranifaciens*.

## 4. Discussion

In the present work, physiological characterisation, including the secretion of extracellular enzymes and biofilm-forming capabilities, was performed for four veil-forming yeast species different from *S. cerevisiae*. These were isolated in a large sampling plan of three different wineries in the Marco de Jerez region where three different Fino wines are produced from the same base wine (i.e., wine var. Palomino Fino, fortified up to 15.5%, *v/v*) [34]. Because these yeasts could be adapted to such stressful conditions during the biological ageing of Sherry wines, it was believed that they could exhibit interesting characteristics to not only carry out vinification, but also as a source of features to improve some technological aspects in winemaking.

The lack of intraspecific diversity found in the present work was initially surprising. Although it is considered that other primers should be used to correctly estimate such diversity, such as the $(GTG)_5$ [37], it should be taken into account, according to the study performed by Esteve-Zarzoso et al. [6], the stage at which a relatively but significantly higher abundance of non-*Saccharomyces* was found was in the sobretablas. Considering therefore, that in this case, the three wineries used the same base wine (the same sobretablas), it seems logical to attribute this low diversity to the fact that the potential source of non-*Saccharomyces* was the same for all three wines.

Despite the low diversity recorded, the biochemical characteristics of the strains analysed have been surprising, even disagreeing with Kurtzman [26] on some issues. Specifically, it was assumed that the *Pichia* genus could not ferment any compound other than glucose, although in this work, others, such as galactose (both in the *P. kudriavzevii* and *P. manshurica* strains), maltose (in the *P. kudriavezvii* strain) and sucrose (in *the P. manshurica* strain) have been found. Both strains were able to assimilate ethanol and nitrite as well. This discrepancy in the biochemical capabilities regarding taxonomy may be due to the very environment in which these yeasts have been exposed to, where interspecific conjugation events may have occurred. Further studies are needed to understand these phenomena, which may be of interest to understand the domestication process undergone in this particular environment. Aerobic growth on ethanol media has been described as a mechanism by which some film-forming strains of the *Pichia* genus may even grow on already fermented beverages [38]. Ethanol tolerance is therefore not surprising in this genus, although the results obtained in the present work reveal a considerably higher degree of resistance than previously reported (i.e., 15% *v/v* in a few strains, usually about 10% *v/v* [39,40]). Here, the strains tested in this study showed a tolerance of above 15.5% ethanol, reaching up 17% *v/v* in the case of *P. manshurica*.

Extracellular enzyme secretion is not typical of a particular genus or species, but depends specifically on the yeast strain [41]. In this case, it was interesting to find β-glucosidase activity not only in the species *W. anomalus* [42], but also in *P. kudriavzevii*. β-glucosidase activity is known to be one of the most important at the oenological level, as it plays a key role in the release of terpenes and other volatile compounds from non-volatile precursors by breaking glycosidic bonds [43,44]. In addition, a weak degree of lipolytic activity was found in *W. anomalus*, being more noticeable in the *P. kudriavzevii* and *P. manshurica* strains. Although lipolytic activity is not essential, it may be able to degrade lipids from grapes or result from yeast autolysis, a common event during biological ageing, thereby releasing free fatty acids and also improving wine quality [45]. Such activity was previously reported in *W. anomalus* [46], but not in the other two species. Regarding proteolytic activity, it is considered to be a key extracellular activity, given that it can allow for wine stabilisation by preventing protein haze, since proteins are hydrolysed into peptides, which in turn can be metabolised by the other yeasts present [8,40]. Our results showed that the *P. manshurica* strain was the only one that possesses a strong proteolytic activity, with it also being the first time that such a property is reported for this species. Moreover, it was also the only yeast tested that showed urease activity, which is of particular interest in this process due to the formation of ethyl carbamate from a spontaneous reaction between the ethanol present in the medium and the urea released during alcoholic fermentation. Hence, by degrading the urea present in the wine during biological ageing, a lower concentration of ethyl carbamate will be formed, with the beneficial health implications that this implies.

The results obtained in the present work suggest domestication events within these species of the genus *Pichia* due to their adaptation to this specific anthropogenic environment. Such a domestication could be consistent with what have been observed in *S. cerevisiae* flor yeasts isolated throughout Europe, which have a common phylogenetic origin and constitute a cluster that is closely related to the clade of the same wine species [47,48], or with that recently observed in the species *Lanchacea thermotolerans* [49]. Furthermore, it is important to note the dynamism of the genus *Pichia*. For instance, *W. anomalus* was formerly *Pichia anomala*, due to its morphological and physiological characteristics, but it subsequently dropped out of the genus due to genetic and phylogenetic characteristics [50], while *Candida krusei* and *P. kudriavzevii* are currently considered as synonyms, and also *Issatchenkia occidentalis* [51].

In this sense, it should be noted that the present study was the first to discover the species *P. manshurica* in this particular system [22], having being thereafter also found in the same system, but in the Montilla-Moriles region (Córdoba, Spain) [52]. In this later work, however, species identification was performed via metabarcoding carried out at an initial regrowth stage (YPD, wine). However, it is necessary to take into account the

limitations that exist nowadays to carry out a correct identification of yeast isolates. Further molecular studies are needed to investigate this possible domestication of the genus due to the biological ageing niche, together with potential implications for the taxonomy of the genus *Pichia*.

In the work carried out in the Montilla-Moriles region [52], authors hypothesised that, although these yeasts seem to be adapted to such stressful and specific conditions, as previously suggested [22], they could not be in a biologically active form. This conjecture was because in their study they did not observe that non-*Saccharomyces* yeasts could form biofilm in the absence of fermentable carbon sources. However, in the present study, the four non-*Saccharomyces* species analysed were able to form biofilm in pure culture and in base wine under winery conditions, as well as in a synthetic media, which only contained ethanol and glycerol as carbon sources. This is consistent with the ability shown of different compounds aerobically assimilated (including ethanol and glycerol) via traditional characterisation methods.

Although we believe that this may have been the beginning of the domestication of this genus to these specific conditions, its activity may reach further. This is demonstrated by the phenotypic characterisation carried out in the present work, which shows its capacity for the aerobic assimilation of different carbon and nitrogen sources, and its ability to secrete extracellular enzymes, as well as its capacity to form a biofilm on its own in the absence of easily assimilable carbon sources. Furthermore, these non-*Saccharomyces* are able to develop an extracellular matrix that stabilises the structure and that serves as communication between the yeast cells.

These results suggest that they can form a biofilm on their own, being metabolically active, and they may even interact with *S. cerevisiae* flor yeasts during biological ageing, providing new distinctive and interesting characteristics in these wines. Further studies are needed to better understand their contribution in wine during biological ageing, as well as other potential applications in the oenological field.

**Author Contributions:** Conceptualisation, G.C.-B., S.M.-V., F.P. and M.R.-M.; methodology, M.R.-M.; software, M.R.-M.; formal analysis, M.R.-M.; investigation, G.C.-B., S.M.-V., F.P. and M.R.-M.; data curation, M.R.-M.; writing—original draft preparation, M.R.-M. and M.H.-F.; writing—review and editing, M.R.-M. and G.C.-B.; visualisation, M.R.-M. and M.H.-F.; supervision, G.C.-B.; project administration, G.C.-B.; funding acquisition, G.C.-B. and J.M.C. All authors have read and agreed to the published version of the manuscript.

**Funding:** This research was co-funded by CDTI, grant number IDI-20180007; and the European Regional Development Fund (ERDF), grant number FEDER-UCA18-106947, within the Intelligent Growth Operational Program with the aim of promoting research technological development and innovation.

**Institutional Review Board Statement:** Not applicable.

**Informed Consent Statement:** Not applicable.

**Data Availability Statement:** Not applicable.

**Acknowledgments:** We are very grateful for the support of Bodegas Lustau and Bodegas Caballero in the sample supply for this study. We also thank the Servicios Centrales de Investigación Científica y Tecnológica (SC-ICYT) of the University of Cádiz, especially Juan González García, for obtaining the electron microscopy photographs.

**Conflicts of Interest:** The authors declare no conflict of interest.

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
