# Peer review of "Non-Saccharomyces Are Also Forming the Veil of Flor in Sherry Wines"

_fermentation, doi:10.3390/fermentation8090456_

Round 1

Reviewer 1 Report

This is a fairly straightforward manuscript, in general not badly written and the discussion put this work nicely in perspective. I do, however, have major issues with the results and how it is presented (as mentioned below). I have put it as a rejection but would strongly encourage the authors to address the points made and resubmit. I am sure there will be a better outcome.

Abstract: The first sentence does not make sense. Make sure about the meaning of “assailable” (Line 12, 37). Not sure one could say that biological ageing is always associated with biofilm formation.

Introduction:

Line 35: should be well-established

Line 179: should be inulin

Results from table 1: I cannot fault the methods, but it goes against what is known about the genus.

I would refer to the instructive article on the Pichia species.

http://dx.doi.org/10.1016/B978-0-444-52149-1.00057-4.

Here they claim things that are in contrast to what is found in Table 1.
The Pichia genus cannot utilize galactose whereas P. manshurica and P. kudriavzevii can ferment it according to the manuscript.

The Pichia genus cannot utilize maltose whereas P. kudriavzevii can ferment it according to the manuscript

Pichia cannot utilize sucrose whereas P. manshurica can ferment it according to the manuscript.

I also find it highly doubtful that an organism will exhibit cellulase activity but not beta-glucosidase activity. Based on the method it would seem like P. membranifaciens and maybe others can actually grow on CMC as sole carbohydrate source but P. membranifaciens showed a halo after congo red staining. Could the cellulase plate assays be included in the manuscript. I do not know if it is in fact cellulase activity or maybe a different reaction that takes place.

Check spelling of “assimilation” and “cellulolytic”

There are thus a lot of uncertain claims made in this table that goes against what is known about this genus. This is also not well discussed. I would suggest omitting the data mentioned above unless it can be substantiated with other means.

Figure 1. I am not a phylogenetic expert but based on http://dx.doi.org/10.1016/B978-0-444-52149-1.00057-4 the LSU (D1D2) sequence is better for resolution of the Pichia genus and it might be why the strains are not grouping very well. If you really want to identify the strains I would recommend concatenate the ITS with the D1D2 region.

Figure 2. This is a mess. I would highly recommend splitting the data in two or more figures. I would also not recommend Excel for drawing graphs for manuscript preparations.

Figure 3. Plate D looks dry and should be redone

Figure 4: I would recommend comparing the structures with yeasts that were grown while shaken and not forming a biofilm/pellicle

Line 320: it is not uncommon for yeast to have proteolytic activity

Author Response

This is a fairly straightforward manuscript, in general not badly written and the discussion put this work nicely in perspective. I do, however, have major issues with the results and how it is presented (as mentioned below). I have put it as a rejection but would strongly encourage the authors to address the points made and resubmit. I am sure there will be a better outcome.

We really thank the reviewer for the critical and comprehensive review of the manuscript. The comments have undoubtedly helped to improve the manuscript, in particular to better focus the results and the discussion of the results. We hope that the reviewer finds it adequate to be considered for publication.

Abstract: The first sentence does not make sense. Make sure about the meaning of “assailable” (Line 12, 37). Not sure one could say that biological ageing is always associated with biofilm formation.

The first sentence of the abstract has been modified to better introduce the manuscript (Lines 12-15). Also, "assailable" has been changed to "assimilable", as we agree that there has been a meaning mistake.

At least in the case of Sherry wines, biological ageing is indeed associated with the formation of this biofilm, which is precisely its main and distinctive characteristic. Besides the changes in the metabolite composition during ageing, it prevents the wine from being in contact with the air, thus giving it the characteristic pale colour of such wines (Fino, Manzanilla).

Introduction:

Line 35: should be well-established

The spelling mistake has been changed, thank you for pointing it out.

Line 179: should be inulin

We are sorry for such orthographic error; it has been corrected.

Results from table 1: I cannot fault the methods, but it goes against what is known about the genus.

I would refer to the instructive article on the Pichia species.

http://dx.doi.org/10.1016/B978-0-444-52149-1.00057-4.

Here they claim things that are in contrast to what is found in Table 1.
The Pichia genus cannot utilize galactose whereas P. manshurica and P. kudriavzevii can ferment it according to the manuscript.

The Pichia genus cannot utilize maltose whereas P. kudriavzevii can ferment it according to the manuscript

Pichia cannot utilize sucrose whereas P. manshurica can ferment it according to the manuscript.

I also find it highly doubtful that an organism will exhibit cellulase activity but not beta-glucosidase activity. Based on the method it would seem like P. membranifaciens and maybe others can actually grow on CMC as sole carbohydrate source but P. membranifaciens showed a halo after congo red staining. Could the cellulase plate assays be included in the manuscript. I do not know if it is in fact cellulase activity or maybe a different reaction that takes place.

We thank the reviewer for pointing this out. We were also struck that it had one activity and not another. Although we do not rule out that it could be due to another reaction, the yellow halo when the plate was stained with congo red was significant in all the experiments performed. However, we thought it might be possible because it is not the first time that a strain belonging to the species P. membranifaciens has shown cellulolytic activity without b-glucosidase (https://doi.org/10.1080/21501203.2020.1837272).

Check spelling of “assimilation” and “cellulolytic”

It has been checked and corrected, thank you for highlighting these typos.

There are thus a lot of uncertain claims made in this table that goes against what is known about this genus. This is also not well discussed. I would suggest omitting the data mentioned above unless it can be substantiated with other means.

We agree with the reviewer that this issue had not been discussed before in the manuscript (now lines 326-336). These activities had indeed not previously been found in the genus, as collected in Kurtzman. These results are, in fact, what made us think about the possibility that the strains found did not belong exactly to the species identified according to both the ITS and D1D2 regions, or that they may have suffered some kind of evolutionary event that could explain the differences found. 

We apologize that this has not been properly discussed. More emphasis has been given to the strange nature of these results, but all tests were repeated in different times and the same results were obtained, so we believe that it should be published and that it can provide relevant information for future studies.

Figure 1. I am not a phylogenetic expert but based on http://dx.doi.org/10.1016/B978-0-444-52149-1.00057-4 the LSU (D1D2) sequence is better for resolution of the Pichia genus and it might be why the strains are not grouping very well. If you really want to identify the strains, I would recommend concatenate the ITS with the D1D2 region.

Rather than identifying to species level, which was done by concatenating the sequence of both regions, the aim of this dendrogram was to highlight what had been observed in the rest of the tests performed, especially regarding the deviation of the P. manshurica strain, which is far from its type species according to the ITS sequence (which, theoretically, is conservative between species).

Figure 2. This is a mess. I would highly recommend splitting the data in two or more figures. I would also not recommend Excel for drawing graphs for manuscript preparations.

We agree with the reviewer, it was not possible to observe the results correctly by presenting them in this way. The figure has been changed to a table (Table 2), in which the mean +/- standard deviation is included together with the corresponding statistic of each experiment.

Figure 3. Plate D looks dry and should be redone

We agree that the plate is slightly dry at the edges; we had not noticed it before. The photo could be modified for a different one, but we believe that the formation of the structure, as well as the difference with respect to the others, is correctly observed in this one.

Figure 4: I would recommend comparing the structures with yeasts that were grown while shaken and not forming a biofilm/pellicle

We do not believe that this is necessary, given that we wanted to observe the biofilm directly under the electron microscope, and the presence (or not) of an extracellular matrix to bind the yeast cells together and stabilise the structure formed, and desired in this case.

Line 320: it is not uncommon for yeast to have proteolytic activity

We are sorry for the digression; this phrase has been deleted as it was not relevant to the discussion.

Reviewer 2 Report

The aim of this work was to identify and characterize native non-Saccharomyces strains isolated in Sherry wines from the Jerez area during biological ageing. The focus of the study is very interesting, considering a topic that can be very important for wine producers of this region.

The paper is well written. In the introduction, I think it is necessary to include a few more comments on the technological interest of non-Saccharomyces species, such as what characteristics they would contribute to Jerez wines.

The analysis procedures are well described. The results were analyzed in detail and the discussion is successful.

Author Response

The aim of this work was to identify and characterize native non-Saccharomyces strains isolated in Sherry wines from the Jerez area during biological ageing. The focus of the study is very interesting, considering a topic that can be very important for wine producers of this region.

The paper is well written. In the introduction, I think it is necessary to include a few more comments on the technological interest of non-Saccharomyces species, such as what characteristics they would contribute to Jerez wines.

The analysis procedures are well described. The results were analyzed in detail and the discussion is successful.

We sincerely appreciate the reviewer's comments on the manuscript. In the introduction, we have expanded on the possible technological characteristics that these strains could offer in the wine sector, not only in this region but also in general for these kind of wines (see lines 57-64). We agree that this puts the findings of this work more into perspective.

Reviewer 3 Report

The paper “Non-Saccharomyces are also forming the veil of flor in Sherry wines” is focused on the characterization of indigenous non-Saccharomyces strains isolated in Sherry wine for veil flor formation   and some metabolic peculiarities, such as enzymatic activities, ethanol resistance, hydrophobicity and others. The topic is of applicative interest and the results report novelty for potential non-Saccharomyces flor strains. The experimental plan is well organized and the results are clearly described and discussed. In general, the article requires a review for the language. For example, the authors in some places report assailable carbon sources, imagine they refer to assimilable carbon sources. 

The methods partly require more detail and clarity. In particular:

1)  The assay of ethanol resistance requires improvement and addition of details, such us the pH value of the medium.

2) The first paragraph of the methods should also contain the indication of the commercial flor strains of S. cerevisiae used as control in the experiments. Therefore, the title of the paragraph must be modified and the description of the commercial flor S. cerevisiae must be included.

3) Another unclear point of the methods concerns the determination of beta-glucosidase activity. The authors refer to the book (Kurtzman, C.; Fell, J.W.; Boekhout, T. Yeasts : A Taxonomic Study; Kurtzman, C., Fell, J.W., Boekhout, T., Eds.; 5th ed.; Elvesier: Saint Louis, MO, USA, 2011), but this reference is very vague. Being a method, it is better to cite a scientific article that reports all the steps of the method, allowing the reproducibility.

In the bibliography the names of the genera and species must be corrected: the species in lowercase and species and genus in italics.

In Figure 2, are the data reported as averages? There is no statistical analysis?

Author Response

The paper “Non-Saccharomyces are also forming the veil of flor in Sherry wines” is focused on the characterization of indigenous non-Saccharomyces strains isolated in Sherry wine for veil flor formation   and some metabolic peculiarities, such as enzymatic activities, ethanol resistance, hydrophobicity and others. The topic is of applicative interest and the results report novelty for potential non-Saccharomyces flor strains. The experimental plan is well organized and the results are clearly described and discussed. In general, the article requires a review for the language. For example, the authors in some places report assailable carbon sources, imagine they refer to assimilable carbon sources. 

Thank you for critically reviewing the manuscript as the reviewer's comments have served to improve it substantially, especially in the methods, where we agree that there were some ambiguities. The changes made following the reviewer's proposals are detailed below.

The methods partly require more detail and clarity. In particular:

1) The assay of ethanol resistance requires improvement and addition of details, such us the pH value of the medium.

Thank you for highlighting this point. Further information on this assay has been added, as well as the pH of the medium (see lines 123-125).

2) The first paragraph of the methods should also contain the indication of the commercial flor strains of S. cerevisiae used as control in the experiments. Therefore, the title of the paragraph must be modified and the description of the commercial flor S. cerevisiae must be included.

Thank you for pointing out that this information was missing. The description of the control yeast strain has been included and the title has been changed accordingly (Lines 101-104).

3) Another unclear point of the methods concerns the determination of beta-glucosidase activity. The authors refer to the book (Kurtzman, C.; Fell, J.W.; Boekhout, T. Yeasts : A Taxonomic Study; Kurtzman, C., Fell, J.W., Boekhout, T., Eds.; 5th ed.; Elvesier: Saint Louis, MO, USA, 2011), but this reference is very vague. Being a method, it is better to cite a scientific article that reports all the steps of the method, allowing the reproducibility.

We regret this ambiguity, the method has been extended and a more detailed citation has been included (see lines 133-134).

In the bibliography the names of the genera and species must be corrected: the species in lowercase and species and genus in italics.

We apologize this typing error, the names of genera and species in the references have been corrected.

In Figure 2, are the data reported as averages? There is no statistical analysis?

Due to the initial way of presenting such results, these were effectively the data averages, but the statistical analysis had not been included. Figure 2 has been replaced by a table (please see Table 2), where the averages, standard deviations and corresponding statistics for each assay have been presented.